# Exploring the Impact of Pharmaceutical Excipient PEG400 on the Pharmacokinetics of Mycophenolic Acid Through In Vitro and In Vivo Experiments

**DOI:** 10.3390/ijms26010072

**Published:** 2024-12-25

**Authors:** Chaoji Li, Min Zhang, Yanni Zhao, Dan Yang, Mei Zhao, Leyuan Shang, Xiaodong Sun, Shuo Zhang, Pengjiao Wang, Xiuli Gao

**Affiliations:** 1State Key Laboratory of Functions and Applications of Medicinal Plants, School of Pharmaceutical Sciences, Guizhou Medical University, Guiyang 550025, China; lcj1294515338@163.com (C.L.); minzhang@gmc.edu.cn (M.Z.); zhaoyanni0523@163.com (Y.Z.); 18311824619@163.com (D.Y.); 15085969665@163.com (M.Z.); 17385484160@163.com (L.S.); sunxiaodong@gmc.edu.cn (X.S.); 2Center of Microbiology and Biochemical Pharmaceutical Engineering, Department of Education of Guizhou, Guiyang 550025, China; 3School of Basic Medical Sciences, Guizhou Medical University, Guiyang 550025, China; 4Experimental Animal Center, Guizhou Medical University, Guiyang 550025, China; shuozhang1015@sina.com

**Keywords:** mycophenolic acid, polyethylene glycol 400, metabolizing enzyme, transport, pharmacokinetics, pharmaceutical excipient–drug interactions

## Abstract

Mycophenolic acid (MPA) is a commonly used immunosuppressant. In the human body, MPA is metabolized into mycophenolic acid 7-O-glucuronide (MPAG) and mycophenolic acid acyl-glucuronide (AcMPAG) mainly through liver glucuronidation, which involves UDP-glucuronosyltransferase (UGTs) and transfer proteins. Research has indicated that the pharmaceutical excipient PEG400 can impact drug processes in the body, potentially affecting the pharmacokinetics of MPA. Due to the narrow therapeutic window of MPA, combination therapy is often used, and PEG400 is widely used in pharmaceutical preparations. Therefore, investigating the pharmacokinetic influence of PEG400 on MPA could offer valuable insights for optimizing MPA’s clinical use. In this study, we examined the impact of a single oral dose of PEG400 on the blood levels of MPA in rats through pharmacokinetic analysis. We also investigated the distribution of MPA in various tissues using mass spectrometry imaging. We explored the potential mechanism by which PEG400 affects the metabolism of MPA using hepatic and intestinal microsomes and the Caco-2 cellular transporter model. Our findings reveal that the overall plasma concentrations of MPA were elevated in rats following the co-administration of PEG400, with the AUC_0-t_ of MPA and its metabolite MPAG increasing by 45.53% and 29.44%, respectively. Mass spectrometry imaging showed increased MPA content in tissues after PEG400 administration, with significant differences in the metabolites observed across different tissues. Microsomal and transport experiments showed that PEG400 accelerated the metabolism of MPA, promoted the uptake of MPA, and inhibited efflux. In conclusion, PEG400 alters the in vivo metabolism of MPA, potentially through the modulation of metabolic enzymes and transport.

## 1. Introduction

As research advances, pharmaceutical excipients are gaining attention for their interactions with drugs. With a focus on metabolizing enzymes and transport, we have explored how excipients interact with medications. Mycophenolate mofetil (MMF), which is a precursor drug of MPA, is commonly prescribed for organ transplant rejection and as an immunomodulator to manage autoimmune conditions like AIDS, lupus nephritis, myasthenia gravis, immune thrombocytopenic purpura, autoimmune hepatitis, IgA nephropathy, and atypical forms of aspergillosis [1,2,3]. MPA is a selective, reversible, and non-competitive inhibitor of inosine phosphoribosyl dehydrogenase (IMPDH), which inhibits the proliferation of T and B cells, limiting the enzymatic steps of purine biosynthesis. Due to its narrow therapeutic index, immunosuppressive regimens have to be optimized, as they may disturb the balance between over-immunosuppression and rejection in transplanted patients if they are not properly individualized. It is, therefore, necessary to explore the factors that may influence MPA blood concentrations.

In the liver and gastrointestinal tract, MPA is metabolized by UGTs into its inactive forms of MPAG and AcMPAG, whose structures are shown in Figure 1 [4]. In the liver, MPA is taken up by hepatocytes, glucuronidated to MPAG, excreted into the bile, and then reconverted back into MPA by interacting with intestinal bacteria [5]. A second peak is found when plasma MPA concentrations are measured, which is usually thought to be due to the enterohepatic cycle.

Pharmaceutical excipients are substances used in the production of pharmaceutical products and prescription formulations, which are not active ingredients but have been deemed safe for use. Recent research has shown that these excipients can impact the absorption, distribution, metabolism, and excretion of drugs by affecting gastrointestinal transporters. This can ultimately affect the bioavailability of drugs and lead to potential adverse reactions. Therefore, it is important to consider the interactions between excipients and active drugs [6,7,8]. The influence of pharmaceutical excipients on pharmacokinetics has received increasing attention [9,10]. The pharmaceutical excipient PEG400 is a polycondensation mixture of water and ethylene oxide, which is commonly used in a variety of pharmaceutical formulations (soft gels, injections, solid lipid nanoparticles, etc.). It has been found that PEG400 affects the pharmacokinetic parameters of other drugs when administered concurrently. Fifty-one previous studies have reported that PEG400 increased the bioavailability of ranitidine in men [11], and previous studies have shown that PEG400 inhibits P-glycoprotein (P-gp) efflux, which promotes the intestinal absorption of celiprolol and improves its bioavailability [12]. In their study, Mudra et al. observed that PEG400 had an inhibitory effect on cytochrome P450 3A4 (CYP3A4), resulting in an increase in the area under the curve (AUC) of midazolam. Furthermore, the production of 1′-hydroxyimidazolam was reduced, leading to an improved concentration profile of midazolam [13]. Consequently, it is crucial to explore the potential influence of PEG400 on the pharmacokinetic and tissue distribution profiles of MPA to inform the proper administration of MPA in clinical practice.

Mass spectrometry imaging can clearly show the spatial location information of various known and unknown molecules in biological samples, and compared with conventional tissue distribution measurements, it can better demonstrate the aggregated portion of molecules as well as the distribution relationship, providing more intuitive experimental results. In certain instances, in vitro cellular studies offer a more precise representation of how PEG400 affects drug absorption than whole-animal experiments. Caco-2 cells, derived from a human colon adenocarcinoma cell line, are regarded as an ideal in vitro model for studying drug absorption processes [14]. Studies have shown that cells can fuse, differentiate, and form a dense monolayer that closely mimics human small intestinal epithelial cells when cultured on a porous polycarbonate permeable membrane. This monolayer demonstrates similar morphology, marker enzyme expression, uptake, transport, and permeability characteristics to small intestinal epithelial cells, effectively mimicking the absorption process of these cells [15,16]. MRP (multidrug resistance-associated protein), P-gp (P-glycoprotein), and BCRP (breast cancer resistance protein) are all found in high levels on Caco-2 cells and play significant roles in regulating drug release [17].

According to our previous study, we hypothesized that PEG400 may influence the processing and distribution of MPA in the body when taken orally by affecting metabolizing enzymes and transporter activity. To test this hypothesis, we developed a rapid and accurate method for measuring the levels of MPA, MPAG, and AcMPAG in plasma, microsomes, and cellular transporters using mass spectrometry imaging. Additionally, we conducted experiments in rats and in vitro studies using hepatic microsomes and transporter cultures to further investigate the impact of PEG400 on the pharmacokinetics and distribution of MPA. Our aim was to enhance our understanding of how PEG400 affects MPA and provide valuable insights for the appropriate use of MPA in clinical settings.

## 2. Results

### 2.1. Effects of PEG400 on the Pharmacokinetics of MPA in Rat Plasma

Following the administration of MPA alone (control group) or in combination with PEG400 (experimental group) in rats, the mean blood concentration–time curves were plotted based on the intra-plasma concentrations of MPA and its metabolites measured at various time points, as depicted in Figure 2. The pharmacokinetic parameters are presented in Table 1. It can be observed from the curves in Figure 2A,B that the concentration of MPA and its metabolites in plasma significantly increased with the addition of PEG400. Regarding the pharmacokinetic parameters, the AUC_0-t_ of MPA and MPAG in the control group were 14.158 ± 4.58 and 62.441 ± 8.551 mg·h/L, respectively, while in the experimental group, they were 22.992 ± 7.538 and 80.827 ± 17.894 mg·h/L, respectively. The AUC_0-t_ of MPA and MPAG in the experimental group increased by approximately 45.53% and 29.44%, respectively, compared to the control group (*p* < 0.05). The C_max_ of MPA and MPAG in the experimental group (4.608 ± 2.176 mg/L and 10.953 ± 2.309 mg/L) was slightly increased relative to that of the control group (2.614 ± 0.778 mg/L and 8.872 ± 0.758 mg/L) after the combined use of PEG400. Moreover, the CL of the experimental group (1002.497 ± 469.077 L/h/kg and 302.556 ± 118.659 L/h/kg) was significantly lower compared to that of the control group for MPA and MPAG (2298.615 ± 859.05 L/h/kg and 401.602 ± 128.411 L/h/kg). We calculated the he values of MPAG/MPA in AUC and Cmax and found that both ratios were reduced by the addition of PEG400, indicating that the effect of PEG400 on the blood concentration of MPA was more pronounced, suggesting that PEG400 may have an effect on the metabolism of AcMPAG, another metabolite of MPA. Additionally, t_1/2_ and T_max_ were also affected by the addition of PEG400. The comparison between the control and experimental groups revealed significant differences in the pharmacokinetic parameters of MPA and its metabolites, indicating that PEG400 can effectively influence the pharmacokinetic characteristics of MPA and its metabolites in vivo.

### 2.2. Effect of PEG400 on the Distribution of MPA in Tissues

By applying the spatial imaging technique, the distribution of MPA and its metabolites in the liver, kidney, spleen, and heart of rats after 6 h of drug administration was explored, and the protonated ions ([M-H]^−^, *m*/*z* 319.1187) and their metabolites ([M-H]^−^, *m*/*z* 495.1508) of MPA in the negative ion mode were obtained by the calculation of the software and the molecular formulae of MPA and its metabolites. The *m*/*z* values of the detected substances were further verified in the software and determined as the measured *m*/*z* of the target substances. In Figure 3, the control groups for liver, kidney, and heart are on the left side, the PEG400 group is on the right side, the control group for spleen is on the upper side, and the PEG400 group is on the lower side. The results show that the addition of PEG400 increased the distribution of MPA in tissues in the liver, kidney, spleen, and heart. Regarding metabolites, we observed significant differences in the liver, where metabolites were significantly increased by the addition of PEG400, whereas the difference was smaller in the kidney compared to the liver, and very little metabolite content was detected in the heart and spleen, with almost no difference. To further verify the differences among the tissues, we performed statistical analysis of their intensities, and the results are shown in the right histograms of Figure 3A,B. The four tissues showed a significant increase in the intensity of MPA in the PEG400 group compared with the control group, whereas the metabolites increased only in the liver and kidney, and the difference in the kidney was much smaller than that in the liver.

### 2.3. Effects of PEG400 on UGTs Enzyme Activity

MPA is a substrate for the in vivo two-phase metabolizing enzymes UGTs, which are metabolized in vivo by glucuronidation, primarily through UGT1A8 and UGT1A9 (with UGT1A10 and UGT2B7 playing minor roles), because UGT1A9 and UGT2B7 are found primarily in the liver, and UGT1A8 and UGT1A10 are expressed mainly in the intestine [3]. Therefore, liver microsomes and intestinal microsomes were prepared separately to reflect the effects of PEG400 on the total UGT enzyme activity in the liver and intestine by measuring the post-incubation concentrations of the two MPA metabolites MPAG and AcMPAG. Before the determination, we explored the effect of incubation time on the concentration of MPAG and AcMPAG in liver microsomes to select the most suitable incubation time; the results are shown in Figure 4. Figure 4A,B show the curves of the concentration of MPAG and AcMPAG over time, respectively. The fitting degree of R2 was greater than 0.99, and it can be seen that between 10 and 90 min, the production of MPA glycoside metabolites MPAG and AcMPAG increased linearly with the incubation time. With practical considerations, 60 min was chosen as the official incubation time. As shown in Figure 5A,B, in the liver microsomes incubated with MPA for 60 min, the concentrations of both MPAG and AcMPAG were elevated with the addition of different concentrations of PEG400 compared to the control. As shown in Figure 5C,D, the concentrations of both MPAG and AcMPAG were also increased in the intestinal microsomes after the addition of PEG400. This suggests that PEG400 increased the activity of UGTs, a two-phase metabolizing enzyme associated with MPA, thereby accelerating the metabolism of MPA in vitro.

### 2.4. Transport Studies in Cell Models

Previous studies have shown that the pharmacokinetics of MPA are affected by intestinal absorption and that the metabolites of MPA and MPAG are reabsorbed into MPA in the colon after enterohepatic circulation. For the study of MPA transport, we chose Caco-2 cells derived from a human colon adenocarcinoma cell line, which has been identified as an ideal in vitro experimental model cell for studying drug uptake mechanisms.

The impact of PEG400 on the transport of MPA was studied using the Caco-2 cell monolayer model. Initially, the effects of MPA and PEG400 on the proliferation of Caco-2 cells were assessed using a colorimetric assay. The results are shown in Figure 6, where the selected concentrations of MPA (40 μM) and PEG400 (2, 4, and 8 mM) did not significantly affect cell proliferation. The integrity of the cell monolayer was confirmed through resistance values and alkaline phosphatase activity. The results of the transport experiments performed in Caco-2 cells are shown in Figure 7, indicating that MPA transport is mediated by the transporter and that the efflux ratio decreases when PEG400 is present at low, medium and high doses. The apparent permeability coefficient of MPA increased in the basolateral–basolateral direction with increasing PEG400 concentrations, indicating enhanced MPA uptake, while it decreased in the basolateral–apical direction, suggesting the inhibition of MPA efflux, and the specific numerical results are shown in Table 2. In summary, PEG400 affects MPA transport by influencing the transporter, thereby affecting MPA uptake and efflux. The effects of PEG400 on transporters intensified with concentration, highlighting a mechanism through which PEG400 impacts the pharmacokinetics of MPA in vivo.

## 3. Discussion

MPA is a widely used medication for autoimmune diseases and the prevention of organ rejection after transplantation. It undergoes degradation by esterase enzymes in the stomach, small intestine, blood, liver, and tissues to form MPA, which is responsible for its pharmacological effects. However, most of the MPA binds to plasma proteins in the body, and only a small portion of the remaining free MPA is therapeutically useful [18,19]. The narrow therapeutic range of MPA requires a detailed examination of the factors that affect blood concentrations to prevent adverse effects. It is important to understand potential drug interactions, minimize serious reactions, and ensure the appropriate clinical use of MPA. Pharmaceutical excipients, which are typically thought to be inactive ingredients in drug formulations, have been found to have physiological and pharmacological effects. Surfactants, stabilizers, and lubricants commonly used as excipients have been shown to impact drug transporters and hepatic enzymes, which can affect drug metabolism in the body. For example, a study showed that the combination of tween-polyethylene glycol and nalbuphine increased the C_max_ and AUC of nalbuphine by inhibiting UGT2B7 activity [20]. Additionally, previous research has indicated that PEG400 has the potential to impact the bioavailability of similar protein substrates [21]. As a result, investigating how PEG400 interacts with the key components of medications and its potential association with changes in the function of metabolic enzymes and biological transport proteins in vivo is a valuable area of study.

Three key processes affect the pharmacokinetics of MPA—the intestinal absorption reaction of MMF, enterohepatic recirculation, and the in vivo metabolism and elimination process of MPA [22], the in vivo metabolism process of MPA as well as the main study sites in this experiment are shown in Figure 8. By comparing the pharmacokinetic parameters of MPA and its metabolites in the presence and absence of PEG400, it was found that PEG400 increased the blood concentration of MPA, facilitated the in vivo metabolism of MPA, and increased the C_max_, T_max_, and AUC_0-t_ of MPA and its metabolite MPAG. The effects of PEG400 on MPA and MPAG were found to be different based on changes in the t1/2 and CL/F of MPA and the t1/2 and AUC_0-t_ of MPAG, and it may promote both MPAG production and the clearance of MPAG from the kidney. It has been reported that MPA produces two metabolites in vivo—MPAG, which is produced in high amounts and is inactive, and AcMPAG, a metabolite that is produced in low amounts and may be pharmacologically active and have toxic effects associated with adverse reactions [23,24,25]. Unfortunately, when testing plasma samples, we did not detect AcMPAG, which may be related to its form of presence or the fact that it was too low in plasma, which is consistent with previous reports in rats [26].

To further illustrate the effects of exploring PEG400 on the in vivo process of MPA, we observed the distribution of MPA and its metabolites in the heart, liver, kidney, and spleen of rats by mass spectrometry imaging. The results show that PEG400 increased the distribution of MPA, while the metabolites were increased only in the liver and kidney. The increase in the liver was significantly higher than that in the kidney. For the large increase in metabolites in the liver, we believe that the effect of PEG400 on the biliary excretion of metabolites is one of the reasons. Regarding biliary excretion, which is an important pathway for MPAG elimination from the liver, we previously showed that PEG400 has an inhibitory effect on the efflux transporter MRP2 [27], which in turn is involved in the biliary excretion process of MPAG and thus in the liver–intestinal cycle, but the effects of other transporters are still unknown and need to be further explored and verified. The small difference in metabolite levels observed in the kidney may be due to the fact that it is primarily excreted in the kidney and PEG400 may facilitate the rate of metabolite excretion, resulting in less pronounced differences in metabolite levels in the kidney than in the liver. The combination of hepatic and renal imaging results and the relevant pharmacokinetic parameters seem to validate the previous suggestions that PEG400 affects MPA and metabolites differently, promoting metabolite production and renal excretion, but the above factors need to be further verified. Metabolites are not as abundant in the spleen and heart, probably because they are not major metabolic and excretory organs. The pharmacokinetic and mass spectrometry imaging results show that PEG400 did increase the blood concentration of MPA, enhanced the efficacy of mycophenolic acid, and altered the in vivo process of MPA.

MPA is metabolized in vivo by the two-phase metabolizing enzymes UGTs (mainly UGT1A8 and UGT1A9, with UGT1A10 and UGT2B7 playing minor roles) [28]. MPA is converted to MPAG (about 90% of the metabolite) by glucuronidation of UGT1A9 in the liver and UGT1A8 in the intestine, and to AcMPAG (about 10% of the metabolite) by UGT1A8 and UGT2B7 [29,30,31]. Therefore, we conducted a preliminary study on the mechanism by which PEG400 affects the pharmacokinetic process of MPA from the perspective of metabolizing enzymes. The results of the in vitro enzyme activity experiments show that PEG400 increased the metabolizing enzyme activity of the overall two-phase metabolizing enzyme UGTs in the liver and intestine, promoted the metabolism of MPA, and increased the concentrations of the metabolite products MPAG and AcMPAG, which is consistent with the results of our previous studies on UGTs (mainly UGT1A9) [27], but of course, for UGT1A8, UGT2B7, and UGT1A10, the exact extent of the effect needs to be further explored. Therefore, changes in metabolizing enzyme activities are one of the reasons PEG400 alters the pharmacokinetic process of MPA.

Mycophenolate mofetil is defatted and hydrolyzed in the stomach after administration to release MPA and absorbed in the stomach and proximal small intestine [32,33]. Due to enterohepatic circulation, the metabolite MPAG is excreted in the liver via hepatocytes into the bile and reconverted into MPA by the action of glucuronidase in the intestinal flora, where it is absorbed into the bloodstream again [34,35,36]. Therefore, we chose Caco-2 cells derived from a human colon adenocarcinoma cell line to construct a transporter model to investigate the effects of PEG400 on MPA transporter. The results show that PEG400 promoted the uptake of MPA, inhibited the efflux of MPA, and decreased the ratio of efflux of MPA. This suggests that the effect on the transporter is also one of the mechanisms by which PEG400 alters the metabolic process of MPA. In conjunction with the changes in the MPA blood levels, the effect of PEG400 on promoting the intestinal absorption of MPA is one of the factors contributing to the increase in MPA blood levels. Our previous study found that PEG400 inhibited some of the efflux transporters (MRP2) and affected the blood concentration of the drug [27]. Therefore, the effects of PEG400 on the transporters related to MPA transport are worthy of further investigation and validation. It is worth considering that the amount of MPA metabolites was not measured in the transporter assay, either because the MPA was administered at a smaller concentration or because the Caco-2 cells themselves contained fewer enzymes that produced metabolites that did not reach the response concentration of the instrument.

Metabolic enzymes and transporter proteins are involved in the absorption and metabolism of a wide range of clinical drugs in the gut and liver and may influence the pharmacokinetic profile of co-administered drugs [37,38]. Studies have shown that PEG400 affects the efficiency of drug absorption and increases intestinal permeability when administered orally [39,40]. Theories on the effects of PEG400 on the bioavailability of MPA include affecting drug-metabolizing enzymes, altering the integrity of cellular membranes, altering the expression of transporter proteins, and affecting absorption and excretion [41,42,43]. Therefore, after discovering that PEG400 alters the pharmacokinetic profile of MPA, we conducted a preliminary study of the mechanism from the point of view of metabolizing enzymes and transport (from the point of view of absorption, and metabolism), and the result was, as expected, that PEG400 affects the relevant metabolizing enzymes and transport, altering the pharmacokinetics of MPA. In this mechanism study, we focused a macroscopic point of view, and did not study in depth which metabolism enzymes and transport proteins play major role. Regarding the measured increase in the concentration of MPA in the blood, we confirmed that PEG400 promoted the absorption of MPA, but the increase in the concentration of the drug was often not caused by a single factor. The other factors that cause increases in the blood concentration of MPA, the effects of PEG400 on the hepatic and intestinal circulation of MPA, and the protein binding rate need to be further explored. Interestingly, in our analyses and summaries, we found that the effects of PEG400 on the in vivo processes of MPA seemed to be different from its effects on metabolites. PEG400 promoted the uptake, accelerated the metabolism, and inhibited the efflux of MPA, whereas it seemed to promote the excretion of MPAGs in the kidneys. The effects of PEG400 on the specific relevant transporters of MPA and its metabolites are still not clear. How they differ from each other and how the excretory processes may behave differently deserves further thought and investigation. Notably, PEG400 can be used as a solubilizing agent to increase the hydrophilic and hydrophobic interactions of hydrophobic drugs through the formation of intramolecular and intermolecular hydrogen bonds by its terminal hydroxyl group, which can play a solubilizing role [44,45]. Hence, PEG400 could potentially act as a solubilizing agent, leading to an increase in the blood concentration of MPA. This finding suggests that PEG400 may enhance oral absorption in vivo. Overall, our study investigated the impact of PEG400 on the pharmacokinetics of MPA and identified some factors contributing to its effects. These results may serve as a reference for excipient–drug interactions involving MPA. Based on the spatial imaging results, further studies of the changes in the two metabolites in different tissues are needed to assess whether the blood concentration of the active metabolite AcMPAG is elevated and to determine whether this elevation increases the risk of adverse reactions.

## 4. Materials and Methods

### 4.1. Materials

Mycophenolic acid (MPA) standard (purity ≥ 98%) was provided by InnoChem (Beijing, China), and mycophenolic acid D3 (IS) standard (purity ≥ 98%) was provided by TARGETMOL (Boston, MA, USA). Mycophenolic acid 7-O-glucuronide (MPAG) standard (purity ≥ 98%) was provided by Yuanye (Shanghai, China), and mycophenolic acid acyl glucuronide (AcMPAG) standard (purity ≥ 98%) was provided by Aladdin (Shanghai, China). PEG400, Hank’s buffer (HBSS), phosphate buffer (PBS), and Tris-HCl buffer (pH = 7.4) were purchased from Solarbio (Beijing Solarbio Bio-139 Technology Co. Ltd., Beijing, China). Merck in Darmstadt, Germany, provided HPLC-grade acetonitrile and methanol. Jet Bio (Guizhou, China) supplied Transwell plates (12-well, 0.4 μm polycarbonate membrane), while Merck provided a Millicell^®^-ERS-2 resistivity meter.

### 4.2. Animals

Male SPF-grade Sprague Dawley rats, weighing 200 ± 25 g, were obtained from Changsha Tianqin Biotechnology Co. Ltd., Changsha, China, with Certificate No. SCXK(HN)2019-0014. All experimental procedures were approved by the Ethics Committee of Guizhou Medical University (Guiyang, China) under approval No. 1901090. The rats were kept in standard laboratory conditions (23 °C ± 2 °C, 50% ± 20% relative humidity, 12 h light/dark cycle) with access to food and water ad libitum, and they were acclimatized for 1 week. Prior to the pharmacokinetic and tissue distribution experiments, the rats were fasted for 12 h with free access to water for approximately 12 h.

### 4.3. Cell Culture System

The cells were grown in 1640 medium with 10% fetal bovine serum, 100 IU/mL penicillin, and 100 μg/mL streptomycin under pH 7.4, 37 °C, and 5% CO_2_ conditions. The passaging of cells was carried out using a 0.25% trypsin-EDTA solution when they reached 80–90% confluence. The Caco-2 used in the experiment was obtained from Shanghai Zhongqiao Xinzhou Biotechnology Co., Shanghai, China.

### 4.4. UPLC-MS/MS Analysis of Sample Concentration

The UPLC-MS/MS system utilized in the study consisted of a Thermo Dionex UltiMate 3000 liquid chromatography system connected to a Thermo TSQ Vantage Triple Quadrupole Mass Spectrometer (Thermo Electron Corporation, San Jose, CA, USA) with positive ion mode electrospray ionization (ESI). The samples were separated on a Waters 164 XBridge BEH C18 column (2.1 × 100 mm) at a constant temperature of 30 °C. The mobile phase consisted of acetonitrile (A) and water with 0.1% formic acid (B). The elution gradient was as follows: it started with 80% B for 0.2 min, decreased to 2% B from 0.2 to 2.4 min, was maintained at 2% B from 2.4 to 3.4 min, increased back to 80% B from 3.4 to 3.5 min, and equilibrated at 80% B until 4.6 min. The flow rate of the mobile phase was 0.3 mL/min, the autosampler temperature was set at 5 °C, the injection volume was 5 μL, and the monitored transitions were m/z 321→207 for MPA, 519→343 for MPAG and AcMPAG, and 324→210 for MPA-d3(IS). Additional mass spectrometry parameters included the sheath gas (ARB) pressure at 20 psi, the auxiliary gas (ARB) pressure at 2 psi, the source voltage at 4.8 KV, the capillary voltage at 2.5 KV, the ion transfer tube temperature at 375 °C, and the nebulization temperature at 275 °C.

### 4.5. In Vivo Pharmacokinetic Study

The MMF was administered to 12 male Sprague-Dawley rats via gavage. The rats were randomly divided into two groups: the MPA + saline group (control) and the MPA + PEG400 group (experimental). The dose of MMF used was 40 mg/kg added to saline or PEG400 at a ratio of drug mass to solvent volume of 1:50 (*w*/*v* = 1:50). Blood samples were collected at various time points (0.5, 1, 1.5, 2, 3, 4, 6, 8, 10, and 12 h) after a single gavage administration. The samples were centrifuged and stored at −80 °C until analysis. After repeated freezing and thawing, 50 μL plasma samples and 200 μL methanol (containing 10 ng/mL mycophenolic acid d-3) was added in a vortex to precipitate the proteins, centrifuged 14,000 r/min for 10 min, and used for UPLC-MS/MS analysis.

### 4.6. AFADESI-MSI Spatial Imaging Analysis of MPA

The AFADESI-MSI platform was used in conjunction with a Q-OT-qIT hybrid mass spectrometer (Thermo Fisher Scientific, San Jose, CA, USA). The AFADESI-MSI setup included an aerosolizer, a syringe pump, stainless steel delivery tubing (3 mm ID, 4 mm OD, 500 mm length), and a vacuum pump. An 80% acetonitrile solution was used as the spray solvent at a flow rate of 0.005 mL/min. The spray voltage was set at −4500 V, spray pressure at 0.65 MPa, pumping flow rate at 80 L/min, and scanning range at *m*/*z* 70–1050 with an automatic gain control (AGC) target of 8 × 10³. Scan spacing was 0.1 mm with a resolution of 70,000 at *m*/*z* 200. MSI experiments were conducted continuously in the x-direction at a scanning speed of 0.2 mm/s and in the y-direction at a traveling speed of 0.2 mm/s. Weekly quality calibration was performed using a commercial calibration solution from Thermo Scientific. Tissue samples were collected from 6 male SD rats, randomly divided into 2 groups of 3 rats each, following the dosing and grouping procedures outlined in Section 4.6. The rats were euthanized 6 h after administration, and tissues (liver, kidney, spleen, and heart) were promptly frozen at −80 °C until MSI analysis. Continuous sagittal thin sections were made with a Leica CM3050S cryostat (Leica Microsystems, Wetzlar, Germany), and the tissues were thawed for one day before sectioning by moving them from −80 °C to−20 °C. The frozen tissues were then fixed on the cryostat at a sectioning temperature of −20 °C with a sectioning thickness of 12 μm. After drying, rectangular positioning frames were drawn for optical scanning.

### 4.7. Preparation of Liver and Intestinal Microsomes

Rat liver and intestine microsomes were prepared using a method involving differential centrifugation, with some modifications based on a previous study [13]. Approximately 3 g of fresh rat liver tissue was taken and cooled with phosphate buffer (pH = 7.4) at 4 °C. The tissue was weighed, washed on ice, dried on filter paper, and then cut into pieces on ice trays at 0 °C. The liver tissue was homogenized in an ice bath at a ratio of 1:3 (*w*/*v* = 1:3) of liver tissue to homogenate volume, and the supernatant was obtained by centrifugation at 9000 rpm, 4 °C for 20 min. The supernatant was further centrifuged at 100,000 rpm for 60 min, the supernatant was discarded, and the precipitate was suspended in PBS buffer with 30% glycerol. The suspension was transferred to 1.5 mL test tubes and frozen at −80 °C. Intestinal microsomes were prepared using the same method as for liver microsomes.

### 4.8. Study of Enzyme Activity

This experiment focused on the impact of PEG400 on the enzymatic activity of UGTs in the liver and intestinal microsomes, as MPA is primarily metabolized by UGTs in vivo. Rat liver microsomes were utilized in this study, divided into three groups, with each group repeated three times. PEG400 was administered at low (2 μM), medium (4 μM), and high (8 μM) doses to intervene and observe the substrate conversion in the incubation system. The same treatment was applied to intestinal microsomes. The probe substrate used for UGTs in both the liver and intestinal microsomes was MPA at concentrations of 10, 20, and 40 μM. The activity of UGTs was assessed by analyzing the effect of PEG400 on the conversion of MPAG and AcMPAG.

### 4.9. Microsomal Incubation System and Treatment

To prepare the incubation system (80 µL), a mixture of magnesium chloride (5 mM, 10 µL), liver microsomes (1 mg/mL, 20 µL), PEG400 (0, 2, 4, 8 µM; 20 µL), and 50 mM Tris-HCl (pH = 7.4) was pre-incubated on ice for 15 min. MPA (10, 20, 40 µM) and 5 mM UDPGA (10 µL) were then added, and the reaction was carried out at 37 °C for 60 min. The reaction was stopped by adding ice-cold methanol (100 µL) containing IS, followed by centrifugation at 4 °C, 16,000 rpm for 15 min. The supernatant (100 µL) was transferred to an autosampler, and 5 µL was injected into the UPLCMS/MS system for analysis. The same procedure was followed for intestinal microsomes.

### 4.10. MPA Transport Studies in Cell Models

A total of 20–30 generations of Caco-2 cells were placed in Transwell plates at a density of 2 × 10^5^ cells per well. The plates had polycarbonate membranes that allowed for permeability. To assess the integrity of the Caco-2 cell monolayer membrane, 0.5 mL of cell-containing culture medium was added to the apical side (AP), and 1.5 mL of blank culture medium was added to the basolateral side (BL). The culture medium was changed every 2 days for the first week, and then daily after the first week until day 21. After 21 days, the supernatant was removed, the cell monolayer was discarded, and rinsed twice with Hank’s Balanced Salt Solution (HBSS) at 37 °C. The Trans-Epithelial Electrical Resistance (TEER) was measured using a resistivity meter to assess the integrity of the cell monolayer membrane. Alkaline phosphatase activity, a marker for the brush border and important during monolayer formation in Caco-2 cells, was also measured. Culture medium was collected from both sides of the cell membrane at 7, 14, and 20 days. Alkaline phosphatase activity was determined using a kit, and optical density values were measured at 520 nm using a Varioskan multimode microplate spectrophotometer from Thermo Fisher Scientific. A cell monolayer was considered suitable for transit experiments when the TEER value was greater than 500 Ω-cm^2^, and there was a significant difference in alkaline phosphatase activity.

(1)
$$TEER=({TEER}_{cell\;monolayer}-{TEER}_{blank})\times A_{area}$$


The resistance of the unit monolayer, known as the TEER unit monolayer, was obtained directly from the Millicell ERS-2 resistivity meter, whereas the TEER blank was the resistance measured when a blank HBSS was inserted. The surface area of the filter membrane was 1.12 cm^2^.

A solution of MPA at a concentration of 40 µM in HBSS was prepared and mixed with varying concentrations of PEG400 (2 mM, 4 mM, and 8 mM). This mixture was added to both the apical (AP) and basolateral (BL) chambers of a Transwell system, with a control group containing no PEG400. For the experiment testing transfer from the apical to basolateral side (AP-BL), 0.5 mL of the sample was added to the AP side as the supply chamber, while 1.5 mL of preheated HBSS at 37 °C was added to the BL side as the receiving chamber. Conversely, for the experiment testing transfer from the basolateral to apical side (BL-AP), 1.5 mL of the sample was added to the BL side as the supply chamber, and 0.5 mL of pre-warmed HBSS was added to the AP side as the receiving chamber. The Transwell system was placed on a 37 °C thermostatic oscillator at 60 r/min. Samples were collected from the receiving chamber and supply chamber at 30, 60, and 90 min, and replaced with an equal amount of preheated blank HBSS. The samples were then precipitated with methanol containing an internal standard, centrifuged, and analyzed using UPLCMS/MS to determine the content. The apparent permeability coefficient (Papp) and externally excreted ratio (ER) were calculated based on the results of the experiments.

The formulae for calculating PAPP and ER for a single layer and a monolayer are as follows:
(2)
$$Papp=\frac{dQ/dT}{AC_{0}}$$

where d*Q*/d*T* (ng/mL·s) is the transit rate (ng/mL·s), *A* is the membrane area (1.12 cm^2^), and *C*_0_ (ng/mL) is the initial concentration of the sample in the supply chamber.

(3)
$$ER={Papp}_{BL-AP}/{Papp}_{AP-BL}$$

where *Papp_BL-AP_* is the apparent permeability coefficient when BL is added to the sample and *Papp_AP-BL_* is the apparent permeability coefficient (AP) when the sample is added on the top side.

### 4.11. Statistical Analysis

All data were presented as the means ± standard deviations (SDs). Statistical analysis of the pharmacokinetic parameters and time series data of the MPA and its metabolites was conducted using DAS 2.0 software. Graphs were generated using GraphPad Prism^®^ 9 (GraphPad software), and differences in means were evaluated for statistical variability using *t*-tests. A significance level of *p* < 0.05 indicated a statistically significant difference, while *p* < 0.01 was considered highly statistically significant.

## 5. Conclusions

To summarize, when pharmaceutical excipient PEG400 is administered concomitantly with MPA, it alters the in vivo metabolic process of MPA. It has been shown that metabolizing enzymes and transport play a role in how PEG400 affects the pharmacokinetics of MPA. However, further research is needed to fully understand the specific mechanism by which PEG400 impacts MPA.

## Figures and Tables

**Figure 1 ijms-26-00072-f001:**
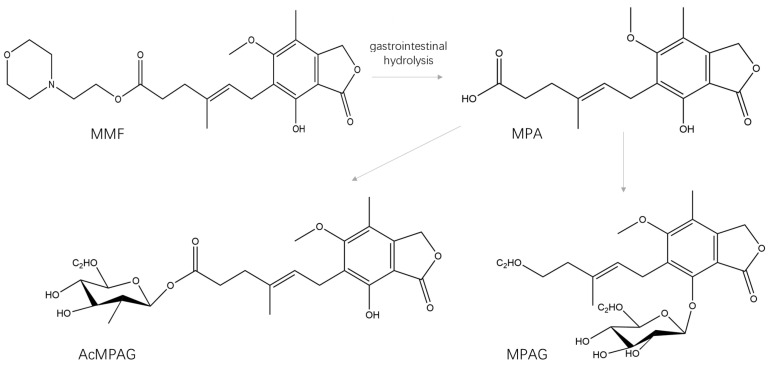
Structural diagram of the metabolism of MMF (mycophenolate mofetil), MPA (mycophenolic acid), MPAG (mycophenolic acid 7-O-glucuronide), and AcMPAG (mycophenolic acid acyl glucuronide).

**Figure 2 ijms-26-00072-f002:**
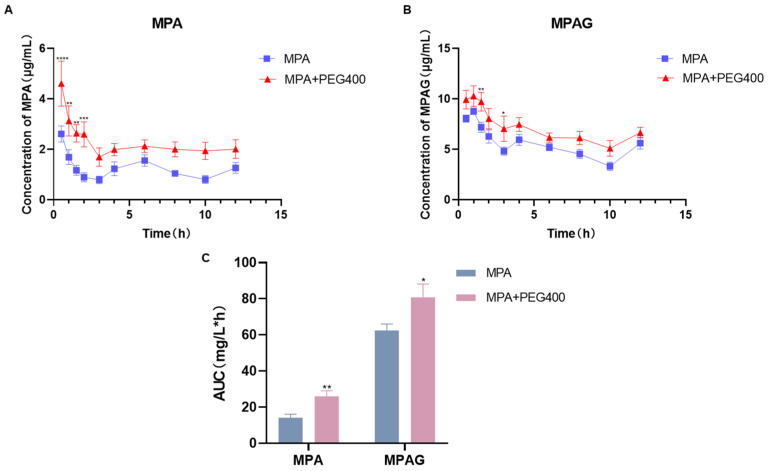
Comparison of MPA blood concentrations in the presence and absence of PEG400. (**A**) Blood concentration–time curve of MPA, (**B**) blood concentration–time curve of MPAG, (**C**) AUC of MPA and MPAG. * *p* < 0.05; ** *p* < 0.01, *** *p* < 0.001; **** *p* < 0.0001, *n* = 6.

**Figure 3 ijms-26-00072-f003:**
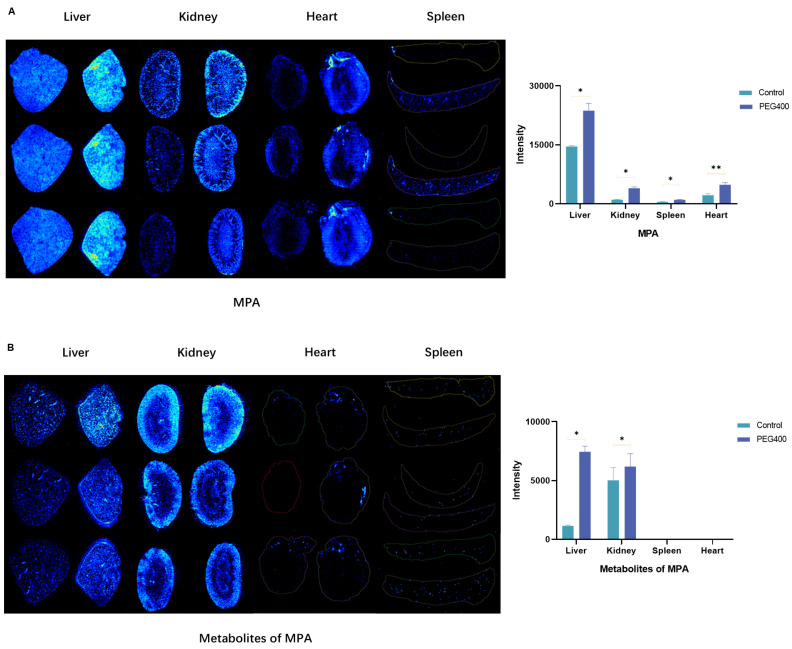
Spatial mass spectrometry imaging results. (**A**) Distribution and intensity results of MPA in the liver, kidney, heart, and spleen. (**B**) Distribution and intensity results of metabolites of MPA in the liver, kidney, heart, and spleen (In the liver, kidney and heart the control group on the left and the PEG400 group on the right, in the spleen the control group on the upper side and the PEG400 group on the lower side). * *p* < 0.05; ** *p* < 0.01, *n* = 3.

**Figure 4 ijms-26-00072-f004:**
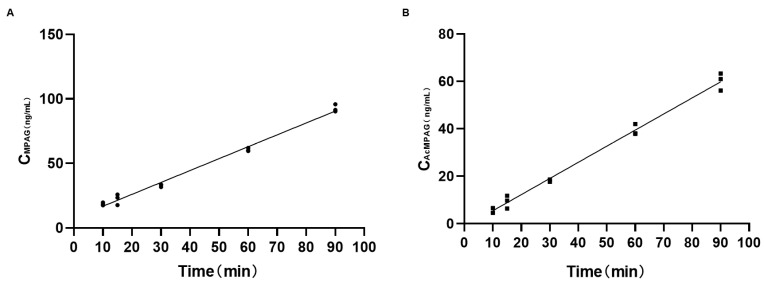
Relationship between incubation time and MPAG and AcMPAG concentrations. (**A**) MPAG concentration over time; (**B**) AcMPAG concentration over time.

**Figure 5 ijms-26-00072-f005:**
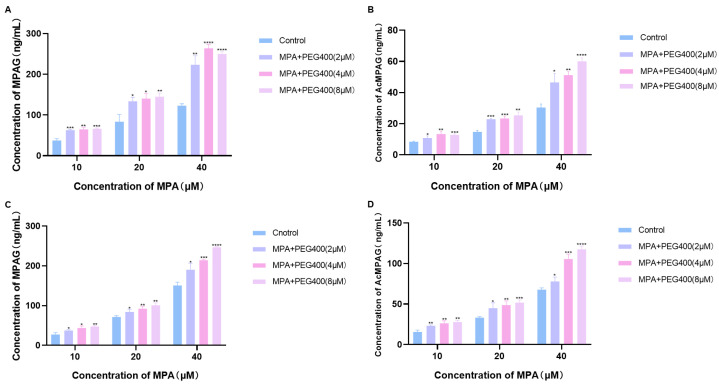
Effects of PEG400 on metabolic enzymes (UGTs) in microsomes. Changes in (**A**) MPAG and (**B**) AcMPAG concentrations after incubation of MPA (10, 20, 40 μM) with different concentrations of PEG400 (0, 2, 4, 8 μM) in liver microsomes. Changes in (**C**) MPAG and (**D**) AcMPAG concentrations after incubation of MPA (10, 20, 40 μM) with different concentrations of PEG400 (0, 2, 4, 8 μM) in intestinal microsomes. * *p* < 0.05; ** *p* < 0.01; *** *p* < 0.001; **** *p* < 0.0001, *n* = 3.

**Figure 6 ijms-26-00072-f006:**
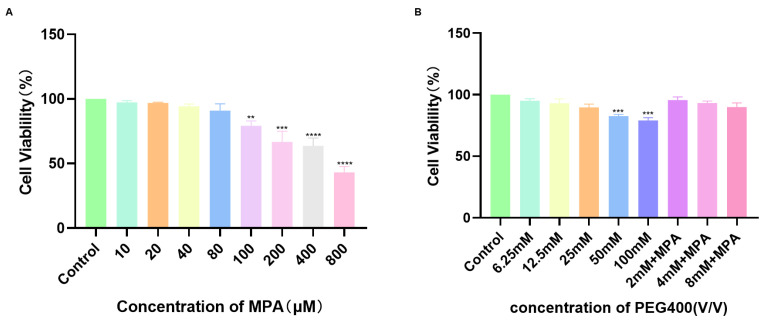
Cell viability. (**A**) Effects of different concentrations of MPA on Caco-2 cell survival; (**B**) Effects of different concentrations of PEG400 on Caco-2 cell survival. ** *p* < 0.01; *** *p* < 0.001; **** *p* < 0.0001. Means ± standard deviations, *n* = 5.

**Figure 7 ijms-26-00072-f007:**
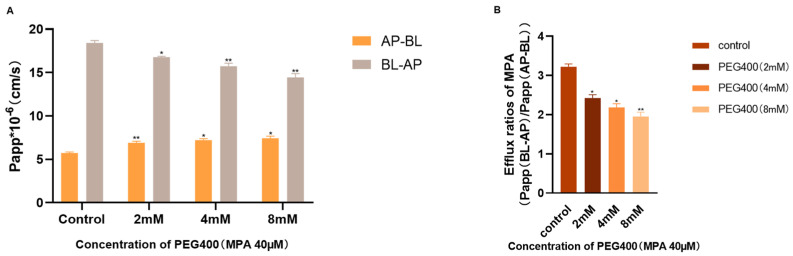
Results of MPA transport by PEG400 in the Caco-2 monolayer epithelial cell model. (**A**) Apparent permeability coefficient Papp values of MPA (40 µM) transport in the Caco-2 cell monolayer model. (**B**) Effects of PEG400 on ER of MPA (40 µM) in Caco-2 cells. * *p* < 0.05; ** *p* < 0.01. Means ± standard deviations, *n* = 3.

**Figure 8 ijms-26-00072-f008:**
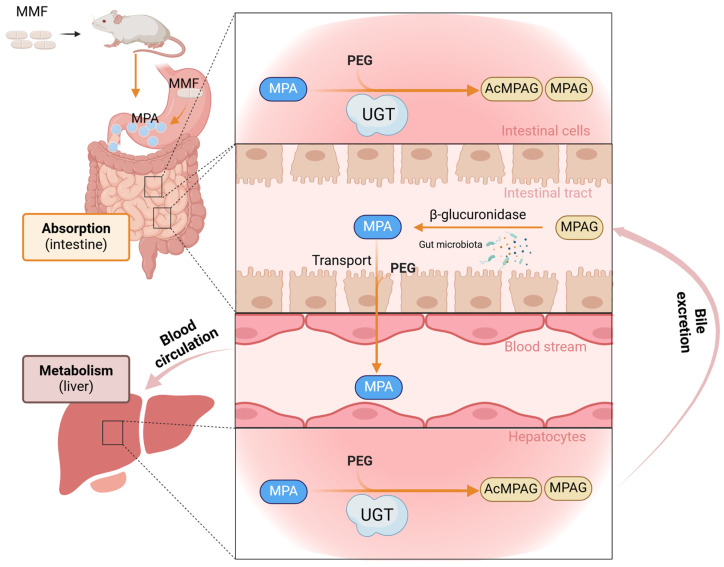
Experimental mechanism diagram. Following oral administration of MMF, MMF is hydrolysed in the gastrointestinal tract to MPA, which is then absorbed into the bloodstream in the intestines.MPA passes through the circulation to the liver, where it is metabolised by the enzyme UGT to MPAG and AcMPAG.A significant amount of MPAG is excreted into the bile, where it is converted by intestinal flora to MPA and reabsorbed into the bloodstream.PEG400 transport affects UGT and transporter proteins, thereby altering the pharmacokinetics of MPA. The transport of PEG400 affects UGT and transporter proteins, thereby altering the pharmacokinetics of MPA.

**Table 1 ijms-26-00072-t001:** Pharmacokinetic parameters of MPA and its metabolites in rats (*n* = 6).

Parameters	Groups	MPA	MPAG	MPAG/MPA
AUC0-t (mg·h/L)	MPA MPA + PEG400	14.158 ± 4.58 25.992 ± 7.538	62.441 ± 8.551 80.827 ± 17.894	4.631 ± 0.889 3.226 ± 0.609
t1/2 (h)	MPA MPA + PEG400	5.261 ± 2.849 11.612 ± 5.899	9.132 ± 3.982 12.308 ± 11.915	/ /
Tmax (h)	MPA MPA + PEG400	0.5 ± 0 0.5 ± 0	0.917 ± 0.204 1.167 ± 0.516	/ /
Cmax (mg/L)	MPA MPA + PEG400	2.614 ± 0.778 4.608 ± 2.176	8.872 ± 0.758 10.953 ± 2.309	3.580 ± 0.766 2.718 ± 0.914
CL/F (L/h/kg)	MPA MPA + PEG400	2.299 ± 0.859 1.003 ± 0.469	/ /	/ /

**Table 2 ijms-26-00072-t002:** Apparent permeability coefficients (means ± SDs, *n* = 3) of MPA on Caco-2 monolayers.

Sample	Group	Papp (×10^−6^ cm/s)		ER
		AP-BL	BL-AP	
MPA	40 uM	5.73 ± 0.19	18.43 ± 0.46	3.22 ± 0.13
	40 uM + PEG400 (2 mM)	6.92 ± 0.32	16.74 ± 0.22	2.43 ± 0.14
	40 uM + PEG400 (4 mM)	7.21 ± 0.28	15.72 ± 0.62	2.18 ± 0.16
	40 uM + PEG400 (8 mM)	7.43 ± 0.44	14.44 ± 0.75	1.95 ± 0.19

## Data Availability

Data are contained within the article.

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
