# Peer review of "Exploring the Impact of Pharmaceutical Excipient PEG400 on the Pharmacokinetics of Mycophenolic Acid Through In Vitro and In Vivo Experiments"

_ijms, 2024, doi:10.3390/ijms26010072_

Round 1

Reviewer 1 Report (Previous Reviewer 2)

Comments and Suggestions for Authors

My main concern is about statistics.

I asked:

1 -4.608 ± 2.176 is different from 2.614 ± 0.778?

2- 10.953 ± 2.309 is different from 8.872 ± 0.758?

3- Which t-test did you run? Paired, unpaired, similar or dissimilar variance?

4- How animals data for any data mean?

5- Please, SHARE RAW DATA

After the examination of data I noticed that Authors run a paired t-test. Paired t-test should be run on data from the same animal before and after treatment (i.e. weight before and after diet). IN this paper, however, you have two groups of independent animals and you MUST run an unpaired t-test. The p-value results = 0.060 for MPA and  0.062 for MPAG, so not significant.

Please revise paper accordingly

Author Response

Reviewer 2 Report (New Reviewer)

Comments and Suggestions for Authors

The authors investigated the effects of the excipient PEG400 on the pharmacokinetics of orally administered MPA (given as MMF) in rat, which resulted in relevant change of plasma PK parameters and tissue distribution of MPA and its major metabolite MPAG. The discussion and interpretation of the results seemed however not quite straight-forward and partially contradictory to the results:

- When coadministered with PEG400, there is an increase of MPA and MPAG. However, the increase of MPAG seems to be not as pronounced as that of MPA (ratio MPA/MPAG reduced). This could be either due to the inhibition of the metabolite formation (inhibition of UGT by PEG400, which would be contradictory to the in vitro results with liver and intestinal microsomes) or an increased secretion of MPAG, which would be contradictory to the kidney distribution results. It is highly recommended to quantify MPA and MPAG in the urine

- The increased liver concentration of MPAG could not be a result of enterohepatic circulation because only MPA is reabsorbed after the degradation of MPAG in the gut. In case of enterohepatic circulation, one would expect a similar increase of MPAG in the liver as for MPA.

- The increased liver concentration of MPAG could be a result of the inhibition of biliary excretion of MPAG. Although the results from Caco-2 experiment suggest an impact of PEG400 on the transporters of MPA, it is not known whether MPA and MPAG would share the same transporters. In addition, transporters involved in the intestine and in the liver might be different

- There is no clear justification for the concentrations of PEG400 used in the in vitro experiments. Are these concentrations (by the way, the concentrations were 2, 4 and 8 mM in text, Figure 6 and Table 2, but 2, 4, and 8 µM in Figure 7) relevant for the in vivo situation?

Round 2

Reviewer 2 Report (New Reviewer)

Comments and Suggestions for Authors

A few minor issues need to be solved:

- Table 1: The term CL/F should be used because this was a oral administration.

- Table 1: CL/F should not be calculated for MPAG because only MPA was dosed. Discussion about CL/F for MPAG should be replaced by the use of AUC of MPAG

- The term exocytosis (multiple places) should not be used for drug transporters, instead efflux should be used.

Author Response

This manuscript is a resubmission of an earlier submission. The following is a list of the peer review reports and author responses from that submission.

Round 1

Reviewer 1 Report

Comments and Suggestions for Authors

I am impressed by the thoroughness of the research and its description demonstrated by the Authors. Each step of the study is precisely presented and discussed. In Table 1 - to assess the effect of PEG400 on drug metabolism, it is worth adding the MPAG/MPA ratio for Cmax and AUC and analysing the values obtained in the context of metabolic changes. There is a lack of explanation of the symbols under many of the figures and tables, which is worth completing.

Reviewer 2 Report

Comments and Suggestions for Authors

Line 278-280: “The Cmax of MPA and MPAG in the experimental group (4.608 ± 2.176 mg/L and 10.953 ± 2.309 mg/L) was significantly higher than that of the control group (2.614 ± 0.778 mg/L and 8.872 ± 0.758 mg/L) after the coadministration of PEG400”

Are you sure 10.953 ± 2.309 mg/L is different from 8.872 ± 0.758 mg/L? Which test did you run?

Figure 2A and 2B: could you run a 2 way ANOVA?

Figure 7: please add the concentration of MPA in the donor compartment

 Due to accumulation of metabolites in many tissues (Fig 3B), I suggest evaluating the effect of transporters also on metabolites

DISCUSSION

Line 396-398: “the majority of MPA binds to proteins in the body, leading to a loss of its pharmacological activity. The small amount of remaining MPA still exerts its therapeutic effect in its free form.” This comment is not very clear to me since no data about protein binding was included in the exp section. Moreover, drug binding to plasma protein is a common feature of many drugs and it is not necessarily represent a problem

Lines 412: why did you included “deglutition”? Which is its role in your experiments since you administer via gavage bypassing deglutition?

MAJOR

IN my opinion entero-hepatic recirculation should be definitely assessed by measuring MPA/MPAG, AcMPAG in bile and feces.

Round 2

Reviewer 2 Report

Comments and Suggestions for Authors

Comments 1: Line 278-280: “The Cmax of MPA and MPAG in the experimental group (4.608 ± 2.176 mg/L and 10.953 ± 2.309 mg/L) was significantly higher than that of the control group (2.614 ± 0.778 mg/L and 8.872 ± 0.758 mg/L) after the coadministration of PEG400” Are you sure 10.953 ± 2.309 mg/L is different from 8.872 ± 0.758 mg/L? Which test did you run?

Response 1: First of all, I am very sorry, maybe my expression is too problematic, resulting in let you have a misunderstanding, I re-applied the software GraphPad to carry out statistical analysis (t-test), the results show that they have a significant difference, P < 0.05 (P = 0.0273), and at the same time, I have made some modifications of my expression.

REPLY: sorry to bother you again but the questions are:

1-       4.608 ± 2.176 is different from 2.614 ± 0.778?

2-       10.953 ± 2.309 is different from 8.872 ± 0.758?

3-       Which t-test did you run? Paired, unpaired, similar or dissimilar variance?

4-       How animals data for any data mean?

5-       Please, SHARE RAW DATA

Comments 2: Figure 2A and 2B: could you run a 2 way ANOVA?

2 Response 2: First of all I found one of my writing mistakes, in the statistical analysis I mostly applied the test of t-tests, which has been modified in the text, then I used 2 ways to analyze Figure 2A and Figure 2B, which has been marked in the graphs and modified in the pictures.

REPLY figure was not modified in the paper and 2-way-ANOVA  was not run

Response 3:

Figure 7: microM is not uM but µM

Comments 4: Due to accumulation of metabolites in many tissues (Fig 3B), I suggest evaluating the effect of transporters also on metabolites

Response 4: My initial idea when performing the transporter experiment with the Caco-2 model was to measure the drug along with the metabolites, unfortunately I did not measure the metabolite concentration while keeping it the same as the microsomal drug concentration, and later on I tried to increase the concentration of the drug up to 80uM to re-measure it, and I did not measure its metabolites either. We considered that there might be too little of the relevant enzymes in the cells to produce metabolites to the instrumental conditions, and if we continued to increase the dose of the drug administered it would affect the cellular activity, so I did not evaluate the metabolites

               REPLY: I mean measuring in vitro as assessed for MPA in cacao2 monolayer

Comments 5: Line 396-398: “the majority of MPA binds to proteins in the body, leading to a loss of its pharmacological activity. The small amount of remaining MPA still exerts its therapeutic effect in its free form.” This comment is not very clear to me since no data about protein binding was included in the exp section. Moreover, drug binding to plasma protein is a common feature of many drugs and 3 it is not necessarily represent a problem

Response 5: Here I just want to express the characteristics of the in vivo process of MPA after absorption and is quoted from the literature, after oral absorption 96%-99% of the MPA is bound to plasma proteins and thus loses its pharmacological activity, and only a small part of the free MPA really plays a role in the pharmacological effect.In addition, there also exists one of my personal thinking, whether PEG400 has an effect on the protein binding rate of MPA, and whether it is a factor to change

REPLY: binding to plasma protein is reversible, so the drug do not loose its effect. Plasma protein act as a reservoir. Please modify text on the paper

Comments 7: MAJOR IN my opinion entero-hepatic recirculation should be definitely assessed by measuring MPA/MPAG, AcMPAG in bile and feces.

Response 7: I apologise for any misunderstanding that may have arisen as a result of my presentation problems, I found elevated concentrations of both MPA and its metabolites in my experiments, microsomal experiments showed that PEG400 activated metabolising enzyme activity, which prompted me to think about elevated blood concentrations of MPA, so I then carried out a transporter experiment, and found that PEG400 facilitated the uptake of MPA but that this facilitation was not sufficient for the actual But this facilitation was not sufficient for the actual increase in blood concentration, so I think this is one of the factors in the increase in MPA concentration.Since there is a entero-hepatic circulation process of MPA, the change of the concentration of MPAG in blood and the results of mass spectrometry imaging drew my attention to the fact that there may be a role of enterohepatic circulation in the elevation of MPA blood concentration caused by PEG400, so the introduction of the enterohepatic circulation was one of my thoughts, and also an outlook on the future research, and in fact, no research was carried out for the evaluation of the enterohepatic circulation. I would also like to thank you very much for the suggestions you have made, which have given us ideas and methods for our subsequent research

REPLY: since you did not demonstrated entero-hepatic circulation you should remove or deeply modify fig8 and all texts related to entero-hepatic circulation
